# Metabolomics and Transcriptomics Analyses Uncover the Potential of Flavonoids in Response to Saline–Alkali Stress in *Codonopsis pilosula*

**DOI:** 10.3390/biology14121759

**Published:** 2025-12-09

**Authors:** Jinhua Liu, Yongqing Wan, Xiaowei Sun, Wenting Su, Kaixia Li

**Affiliations:** 1School of Basic Medicine, Changzhi Medical College, Changzhi 046000, China; liujinhua@czmc.edu.cn (J.L.);; 2Inner Mongolia Key Laboratory of Plants Adversity Adaptation and Genetic Improvement in Cold and Arid Regions, Inner Mongolia Agricultural University, Hohhot 010018, China

**Keywords:** metabolomics, transcriptomics, flavonoids, saline–alkali stress, *Codonopsis pilosula*

## Abstract

*Codonopsis pilosula* (Cp) is a cherished traditional Chinese medicinal herb, holding significant value in disease therapy and nutritional wellness. Nevertheless, soil salinization poses a severe impediment to the extensive cultivation of Cp, underscoring the pressing need to unravel its saline–alkali response mechanisms. This pioneering study delves into how the flavonoids within Cp react to saline–alkali stress. Initially, metabolomic profiling was employed to pinpoint distinct flavonoids of Cp under saline–alkali stress. Subsequently, transcriptomic scrutiny was undertaken to identify differential flavonoid biosynthesis-related genes and bHLH transcription factors. Ultimately, a correlational analysis of these two sets of differential genes was performed to preliminarily clarify the saline–alkali response mechanism in Cp. This provides a basis for the development of active ingredients in Cp and its large-scale cultivation.

## 1. Introduction

Currently, global environmental change has become an established fact, posing a threat to the cultivation and planting of medicinal plants [1]. The escalation of global temperatures, coupled with human activities encompassing unsustainable agricultural practices and inappropriate irrigation techniques, has directly triggered a rise in sea levels, which, as a result, has led to an elevation in groundwater salinity across most regions [1]. Moreover, severe salt stress often occurs in conjunction with high pH rather than independently, and the synergistic effects of these two stresses can have a more severe impact on the normal growth of plants [2]. Most regions in China suffer from soil salinization issues, which restrict the planting range of traditional Chinese medicinal herbs such as *Codonopsis pilosula* (Cp). Therefore, investigating the effects of saline–alkali stress in Cp and its potential coping mechanisms will facilitate the successful cultivation of this herb on saline–alkali land, thereby expanding its planting areas.

Flavonoids are an important class of phenolic secondary metabolites widely distributed in the fruits, roots, stems, leaves, and flowers of most plants. They are complex mixtures composed of multiple components, including flavanols, flavanones, chalcones, anthocyanins, and isoflavones [3]. Flavonoids have significant active ingredients that provide metabolic regulation, antioxidant, anti-cancer, hypoglycemic, and lipid-lowering effects, making them essential nutrients for human health [4]. In addition, flavonoids, as plant-derived secondary metabolites, not only contribute to pigmentation and flavor formation but also function as antioxidants by scavenging reactive oxygen species (ROS), thereby mitigating abiotic stresses (e.g., UV radiation, low temperature, and salinity–alkalinity) and acting as key mediators in plant environmental adaptation strategies [5,6,7,8].

The biosynthesis of flavonoids is mainly carried out through the phenylpropane metabolic pathway, in which the genes encoding the relevant enzymes are regulated by various transcription factors. The expression of genes involved in flavonoid biosynthesis is directly influenced by external environmental factors. For example, RcWRKY transcription factors are associated with differential metabolites of flavonoids in response to UV-B stress [9]; StWRKY41 enhances freeze tolerance by enhancing the antioxidant capacity of flavonoids [10]; and MsMYB12 can directly bind to the promoter of *MsFLS13* and promote its expression, thereby enhancing the accumulation of flavonols and antioxidant capacity to improve the comprehensive tolerance to cold and saline–alkali stress [11]. In summary, flavonoids help plants cope with various abiotic stresses by enhancing their antioxidant activity, and the synthesis of flavonoids is regulated by multiple transcription factors.

Cp is a herbaceous plant belonging to the Campanulaceae family and the Codonopsis genus. It is mainly distributed in the southeastern part of Shanxi Province and is one of the important medicinal and edible Chinese medicinal materials. According to traditional Chinese medicine records, Cp has the effects of harmonizing the spleen–stomach, clearing the lungs, and nourishing blood. Modern medical research has shown that it has the effects of anti-fatigue, enhancing the function of the reticuloendothelial system, improving hematopoietic function, and regulating cardiovascular and cerebrovascular systems, the central nervous system, and gastrointestinal function [12,13]. Cp is mainly used as medicine after drying its root tubers, and its functional components include sugars, alkaloids, flavonoids, etc. [14]. Therefore, Cp has high medicinal, nutritional, and economic value, but its cultivation and expansion are limited by environmental factors.

Given that Cp boasts high medicinal value while its growth is constrained by environmental factors such as salinity and alkalinity, and considering the current severe lack of research on Cp’s response to saline–alkali stress, it is particularly crucial to explore the research mechanism of how Cp responds to such stress through flavonoid metabolites [15]. In this study, firstly, the saline–alkali stress treatments were carried out in Cp, the phenotype was observed, and root, stem, and leaf samples were collected. Secondly, physiological and biochemical indicators, as well as metabolome and transcriptome analysis, were conducted on the collected samples to preliminarily explore the mechanism of saline–alkali stress in Cp. Finally, by combining transcriptome and metabolome analysis, the saline–alkali response mechanism of Cp was further elucidated. In summary, our study preliminarily unveiled the mechanism by which flavonoids in Cp respond to saline–alkali stress, thereby laying a theoretical foundation for the development of flavonoids from different organs, as well as for molecular breeding and the expansion of cultivation area.

## 2. Materials and Methods

### 2.1. Plant Materials, Growth Conditions, and Saline–Alkali Stress Treatments

Cp seeds were purchased from Pingshun County, Changzhi City, Shanxi Province. For seed germination, the seeds were first immersed in warm water (40–50 °C) and continuously stirred until the water temperature naturally decreased to room temperature (15–20 °C), followed by a 5 min soak. Subsequently, the seeds were transferred to a sterile gauze pouch and rinsed thoroughly with distilled water to remove surface mucilage. The cleaned seeds were then sown onto a mixed growth substrate (vermiculite–nutrient soil, 4:1, *v*/*v*), covered with plastic film for moisture retention, and cultured in a controlled greenhouse under long-day conditions (16 h light/8 h dark). The seedlings were transplanted after approximately 2 weeks of growth.

Two-month-old growth-uniform Cp seedlings were selected for saline and saline–alkali stress treatment, with H_2_O (CK), 100 mM NaCl, 200 mM NaCl, 100 mM NaHCO_3_, and 200 mM NaHCO_3_ for 6 days. After processing for 3 days and 6 days, the roots, stems, and leaves were photographed, observed, and sampled. The samples were cleaned with distilled water, wiped dry with filter paper, and quickly frozen in liquid nitrogen before being transferred to a −80 °C freezer for storage. Cp seedlings (roots, stems, and leaves) were subjected to saline–alkali stress for 6 days for physiological and metabolomic tests, and for 3 days for transcriptome sequencing and RNA extraction.

### 2.2. Phenotypic Measurement

For the measurement of plant height, root length, and stem length, CK, NaCl, and NaHCO_3_-treated samples were obtained from three biological replicates, and each biological replicate contained at least three plants. The above indicators were all measured with a ruler.

### 2.3. Analysis of Physiological and Biochemical Indices

The antioxidant indices were all detected using reagent kits (all purchased from Soleibao Technology Co., Ltd., Beijing, China), and the specific steps referred to the instructions of the superoxide dismutase (SOD) activity detection kit (BC5165), the malondialdehyde (MDA) content detection kit (BC0025), the reduced glutathione (GSH) content detection kit (BC1175), the hydrogen peroxide (H_2_O_2_) content detection kit (BC3595), the peroxidase (POD) activity detection kit (BC0090), and the catalase (CAT) activity detection kit (BC0205).

### 2.4. Metabolome Analysis

Lianchuan Biotechnology Co., Ltd. (Hangzhou, China) conducted untargeted metabolome profiling using 54 independent plant samples, with six biological replicates per treatment group for metabolomic analysis. Metabolite extraction was performed using 80% ice-cold methanol, followed by ultra-performance liquid chromatography–high-resolution mass spectrometry (UPLC-HRMS) detection. This approach enabled unbiased detection of all detectable metabolites in the samples, including flavonoids. Subsequently, secondary mass spectrometry (MS/MS) data of the sample metabolites were matched against an in-house MS/MS spectral library, yielding high-confidence metabolite identification results. Data acquisition was carried out using a high-resolution Triple TOF-6600 mass spectrometer (SCIEX, Framingham, MA, USA). Finally, XCMS 1.24.1 software performed preprocessing on the collected mass spectrometry data, including peak picking, peak grouping, retention time correction, secondary peak grouping, isotope and adduct labeling, etc.

The correlation analysis of QC samples (each QC sample was made by mixing equal amounts of 54 prepared metabolomics test samples) was conducted using the Pearson correlation coefficient of R 3.6.3 packager, and the three conditions of *p* < 0.05, difference multiple > 1.2, and VIP calculated by PLSDA analysis obtained by *t*-test were met simultaneously to screen out the final significantly differentially accumulated metabolites (DAMs). Heatmaps, KEGG, and GO maps were generated for the selected DAMs to analyze their expression patterns and pathway enrichment. Correlation analysis between DAMs and their enriched pathways was subsequently performed. All data visualizations were created using the Lianchuan Biological Cloud platform tool 3.6, with metabolomics experiments replicated six times for each treatment.

### 2.5. Transcriptome Analysis and RT-qPCR Identification

Lianchuan Biotechnology Co., Ltd. completed the transcriptome sequencing, utilizing 27 samples, with three biological replicates per treatment group for transcriptomic analysis. Total RNA was extracted using the TRIzol (Thermo Fisher, Waltham, MA, USA, 15596018) method. The quality of RNA was analyzed using the Bioanalyzer 2100 and RNA 6000 Nano LabChip Kit (Agilent, Santa Clara, CA, USA, 5067-1511), and a sequencing library was constructed using high-quality RNA samples with RIN values > 7.0 and an OD260/280 value of 1.8–2.1. Sequencing was performed using the Illumina NovaseqTM 6000 platform (USA). Differential expression analysis between the treatment group and the CK group was conducted using DESeq2. The threshold criterion for screening differentially expressed genes (DEGs) was fold change ≥ 2. A Venn diagram was constructed to identify the overlapping DEGs among multiple experimental groups. Heatmaps were generated to visualize expression patterns, network diagrams were constructed to explore correlations, and KEGG pathway diagrams were plotted to identify enrichment pathways of the selected DEGs. The above data were plotted using the Lianchuan Biological Cloud platform tool. The transcriptome experiments were replicated 3 times for each treatment.

The extracted RNA was reverse-transcribed into cDNA and diluted 10 times as a template for real-time fluorescence quantitative PCR. The reverse transcription (RT mix with DNase) and fluorescence quantification reagents (Universal SYBR Green qPCR Supemix) were sourced from Suzhou Youyilandi Biotechnology Co., Ltd., Suzhou, China, and users should consult the instructions for detailed operating steps. The fluorescence quantitative instrument was Archimed Analyzer V1.0, and the data was exported and calculated using the 2^−ΔΔCt^ method. Each gene had three biological replicates.

### 2.6. Data Analysis

The data were expressed as mean ± standard error (SEM). All data were processed and underwent statistical analysis using Excel 2019 and IBM SPSS Statistics V20.0 for one-way analysis of variance (ANOVA), followed by Duncan’s multiple range tests (*p* < 0.05).

## 3. Results

### 3.1. Phenotypic Observation of Cp Under Saline–Alkali Stress

To investigate the effects of saline–alkali stress on Cp seedlings, plants were cultivated in a greenhouse for two months, after which uniformly grown seedlings were selected for treatment. Seedlings were subsequently exposed to saline (100 mM or 200 mM NaCl) or saline–alkali (100 mM or 200 mM NaHCO_3_) solutions. On the 3rd and 6th days post-treatment, experiments including photography, physiological measurements, and organ sampling were conducted (Figure 1A). After, different organ samples were collected for data determination. The measurement results showed that the plant height of the treated groups was lower after the early treatment of NaCl or NaHCO_3_. Further observation found that early treatment of NaCl had a greater impact on the aboveground part, but did not produce a noticeable effect on the underground part of Cp. The short-term NaHCO_3_ treatment had an obvious inhibitory effect on both the aboveground and underground parts of Cp (Figure 1). The comparison results between salt stress and saline–alkali stress mentioned above indicated that saline–alkali stress had a more pronounced inhibitory effect on the aboveground and underground growth of Cp, suggesting a synergistic inhibitory effect of saline–alkali conditions on the plant height of Cp. 

### 3.2. Physiological Indices: Detection of Different Cp Organs Under Saline–Alkali Stress

In order to investigate the antioxidant mechanism of Cp under saline–alkali stress, antioxidant indices (SOD, MDA, GSH, H_2_O_2_, POD, and CAT) were tested on root, stem, and leaf samples treated with 100 mM NaCl and NaHCO_3_ for 6 days. The results showed that the antioxidant indices of different organs changed differently at the same concentration. For example, compared with the CK samples, the H_2_O_2_ content in roots and stems increased while it decreased in leaves, indicating that appropriate saline–alkali stress was beneficial for the clearance of H_2_O_2_ in leaves. In addition, SOD is an enzyme that can catalyze the dismutation of superoxide anion radicals (O^2−^) into H_2_O_2_ and O_2_. The low H_2_O_2_ content in leaves might also be due to the lower activity of this enzyme in leaves after stress treatment. In addition, compared with the CK group, the content of MDA (a key product of lipid peroxidation in organisms) in roots and leaves increased while it decreased in stems, indicating that appropriate saline–alkali stress was beneficial for clearing lipid peroxidation products in stems, while GSH, which had antioxidant activity, changed in the opposite direction and had a higher content in stems. In summary, for the same concentration of saline–alkali stress, different organs might have different antioxidant mechanisms (Figure 2). Consequently, transcriptome and metabolomic tests were conducted on samples from various organs to investigate the saline–alkali response mechanism of Cp.

### 3.3. Metabolome Analysis of Different Organs of Cp Under Saline–Alkali Stress

To explore Cp’s metabolic response to saline–alkali stress, a 6-day stress treatment was applied, followed by metabolomic tests of root, stem, and leaf samples. Firstly, the Pearson correlation coefficient was employed to analyze QC sample abundance post-quality control, with a coefficient close to 1 indicating high correlation, good repeatability, and stable instrument performance, and in this study, the high correlation rendered the samples suitable for subsequent sequencing (Figure 3A). Secondly, in the PLS-DA score plot where PC1 and PC2, respectively, represent the first and second principal components with each point corresponding to a sample and closer points signifying more similar expression patterns, the PLS-DA analysis of flavonoids in this experiment revealed that PC1 and PC2 accounted for 8.18% and 66.53% of the variance, respectively, and the plot exhibited clustered repeated samples along with good organs and treatment distinction (especially among organs), indicating high repeatability within groups and effective group separation across the nine groups (Figure 3B). Additionally, the permutation test plot, which involved randomly shuffling sample labels for modeling and prediction to generate R^2^ and Q^2^ values, showed that after 200 shuffles, when regression lines (red for R^2^ and blue for Q^2^) were plotted, the model in this study met the criteria of having the R^2^ line above the Q^2^ line and the Q^2^ line intersecting the *Y*-axis below 0 within the [0, 1] range, thus indicating its applicability and absence of over-fitting (Figure 3C).

Through untargeted metabolomics, all detectable metabolites in the sample, including flavonoids, were unbiasedly detected. The analysis of the Cp metabolome obtained a total of 725 secondary metabolites (MS2), including 32 flavonoids, which were divided into 7 categories (Figure 3D and Appendix A). There were 23 DAMs of flavonoids in the CK_LeafVSCK_Root group, 18 in the CK_StemVSCK_Root group, and 19 in the CK_LeafVSCK_Stem group. Additionally, in the roots, 9 DAMs of flavonoids were significantly different under NaCl treatment compared to the CK group, and 14 were significantly different under NaHCO_3_ treatment; in the stems, 11 DAMs were significantly different under NaCl treatment and 10 under NaHCO_3_ treatment; and in the leaves, 7 DAMs were significantly different under NaCl treatment and 11 under NaHCO_3_ treatment. Overall, there were differences in the number of DAMs of flavonoids in different groups, which also indicated that different treatments had an effect on the accumulation of flavonoids in different organs of Cp (Figure 3E–H).

The heatmap analysis of DAMs of flavonoids in nine groups revealed notable variations in the compositions and accumulations, indicating that organ type and treatment conditions significantly influenced the relative flavonoid content. As shown in Figure 4, there were 23 DAMs of flavonoids (21 up and 2 down) in CK_LeafVSCK_Root, with Galangin upregulated by 155.99 times and 7-hydroxy-2-(4-hydroxyphenyl)-5-[(2S,3R,4S,5S,6R)-3,4,5-trihydroxy-6-(hydroxymethyl)oxan-2-yl]oxy-2,3-dihydrochromen-4-one downregulated by 0.17 times; 18 flavonoids (18 up) in CK_StemVSCK_Root, among which Cyanidin 3-glucoside cation was upregulated by 44.43 times; and 19 flavonoids (16 up and 3 down) in CK_LeafVSCK_Stem, with 16-Panasenoside upregulated by 17.14 times and 5,7-dihydroxy-2-(4-hydroxyphenyl)-3-methoxy-4H-chromen-4-one and Isorhamnetin downregulated by 0.28 times (Figure 4A–C).

Under NaCl treatment, there were nine flavonoids (one up and eight down) in the root, where Apigenin 7-O-glucoside was downregulated by 0.29 times; 11 flavonoids (10 up and 1 down) in the stem, with Cyanidin 3-rutinoside upregulated by 3.55 times; and 10 flavonoids (7 up and 3 down) in leaves, among which Diosmetin was upregulated by 2.57 times. Under NaHCO_3_ treatment, there were 14 flavonoids (3 up and 11 down) in the root, with Hieracin upregulated by 0.17 times; 10 flavonoids (4 up and 6 down) in the stem, where Cyanidin 3-rutinoside was downregulated by 0.40 times; and 11 flavonoids (10 up and 1 down) in leaves, with Naringenin upregulated by 8.59 times. The above results indicated that different organs under saline–alkali stress might have different flavonoid response mechanisms. Moreover, after NaHCO_3_ treatment, the DAMs related to flavonoids in roots were evidently greater than those observed under NaCl treatment (Figure 4D–I).

KEGG enrichment analysis showed that there were eight flavonoid synthesis-related metabolic pathways in nine groups, including flavonoid biosynthesis (ko00941), anthocyanin biosynthesis (ko00942), isoflavonoid biosynthesis (ko00943), flavone and flavonol biosynthesis (ko00944), degradation of flavonoids (ko00946), biosynthesis of phenylpropanoids (ko01061), biosynthesis of secondary metabolites (ko01110), and metabolic pathways (ko01100). Among them, the ko00944 pathway accumulated the most metabolites (Figure 5A–I). Additionally, metabolites and pathway correlation network analysis showed that these DAMs of flavonoids were significantly correlated with flavonoid synthesis-related metabolic pathways, and the metabolites in all nine groups were significantly associated with the ko00944 pathway (Figure 6A–I).

### 3.4. Transcriptome Analysis of Different Organs of Cp Under Saline–Alkali Stress

Since Cp responded to saline–alkali at the transcriptional level earlier than the accumulation of metabolites, samples treated for 3 days were selected and subjected to transcriptome sequencing, followed by data analysis of root, stem, and leaf samples of Cp to elucidate the mechanisms underlying its transcriptional-level response to saline–alkali stress. Pearson correlation coefficient analysis showed that samples with color gradients closer to red (indicating correlation coefficients near 1) exhibited stronger positive correlations, whereas those closer to white (indicating coefficients near 0) showed weaker correlations. In the graph in Figure 7A, it is evident that samples within the same group exhibited high correlation, approaching red, whereas the correlation between root and leaf samples (including those treated with saline–alkali) was relatively low, approaching white.

PCA analysis of the DEGs revealed that PC1 accounted for 66.1% of the variance, while PC2 accounted for 22.95% of the variance. In the PCA plot, it was observed that duplicate samples clustered together, and there was good discrimination among different organs and treated samples (particularly in the leaves). These findings indicated that the repeatability within these sample groups was very high, and the discrimination between groups was relatively good across the nine groups (Figure 7B).

The thresholds |log_2_FC| ≥ 1 and *p* < 0.05 were established to screen for DEGs. A total of 9364 DEGs were identified in the CK_LeafVSCK_Root groups (4527 up and 4970 down), 8093 DEGs in CK_StemVSCK_Root (4337 up and 3756 down), 6275 DEGs in CK_LeafVSCK_Stem (3380 up and 2895 down), 3319 DEGs in NaCl_RootVSCK_Root (1378 up and 1941 down), 2341 DEGs in NaCl_StemVSCK_Stem (940 up and 1401 down), 759 DEGs in NaCl_LeafVSCK_Leaf (657 up and 102 down), 2236 DEGs in NaHCO_3___RootVSCK_Root (810 up and 1426 down), 2245 DEGs in NaHCO_3__StemVSCK_Stem (1042 up and 1203 down), and 2303 DEGs in NaHCO_3__LeafVSCK_Leaf (1153 up and 1150 down) (Figure 7C). These results indicated that saline–alkali stress had significantly impacted the gene expression profile of PG.

To validate the reliability of the transcriptome dataset, eight DEGs were randomly selected for RT-qPCR quantification. The results of RT-qPCR validation showed the expression patterns of the selected genes were largely consistent with the trends observed in the transcriptome data (Appendix A), thereby confirming the reliability of the transcriptome dataset and supporting its application in subsequent analyses.

KEGG enrichment analysis of DEGs revealed some common pathways with DAMs of flavonoids, including flavonoid biosynthesis (map00941), anthocyanin biosynthesis (map00942), flavone and flavonol biosynthesis (map00944), and so on (Figure 8 and Appendix A). These findings collectively suggest a coordinated transcriptional and metabolic regulation of flavonoid synthesis.

Then, the heatmap analysis of flavonoid pathway genes across different groups revealed significant intergroup variations in gene types and quantities, aligning with metabolic profiling results and further suggesting that saline–alkali stress in different organs might have modulated flavonoid synthesis via distinct genetic pathways. As shown in Figure 8A, there were 54 genes (22 up and 32 down) in CK_LeafVSCK_Root, 48 genes (19 up and 29 down) in CK_StemVSCK_Root, and 33 genes (13 up and 20 down) in CK_LeafVSCK_Stem. For NaCl treatment, there were 15 genes (6 up and 9 down) in NaCl_RootVSCK_Root, 14 genes (7 up and 7 down) in NaCl_StemVSCK_Stem, and four genes (three up and one down) in NaCl_LeafVSCK_Leaf. For NaHCO_3_ treatment, there were nine genes (three up and six down) in NaHCO_3__RootVSCK_Root, 17 genes (8 up and 9 down) in NaHCO_3__StemVSCK_Stem, and 20 genes (8 up and 11 down) in NaHCO_3__LeafVSCK_Leaf. The above results also indicated that different organs under saline–alkali treatments had different flavonoid regulatory mechanisms.

Taking the flavonoid biosynthesis pathway (map00941) as an example, compared to leaves, *Cpi08G002482*-chalcone isomerase 2, *Cpi01G011650*-chalcone synthase, *Cpi02G004415*-chalcone synthase 2, *Cpi02G003147*-flavonoid 3′-monooxygenase-like, and so on, were regulated in roots; compared to stems, *Cpi08G004947*-anthocyanidin reductase and *Cpi08G001765*-anthocyanidin reductase, and so on, were regulated in roots; compared to leaves, *Cpi08G002482*-chalcone isomerase 2, *Cpi03G000419*-chalcone isomerase 3, *Cpi03G002695*-flavanone 3-hydroxylase, *Cpi02G012095*-anthocyanidin synthase, and so on, were regulated in stems; under NaCl treatment, *Cpi01G011650*-chalcone synthase and other genes were regulated in roots, *Cpi08G004947*-anthocyanidin reductase and other genes were regulated in stems, and *Cpi06G004680*-flavonoid-3′,5′-hydroxylase and other genes were regulated in leaves; under NaHCO_3_ treatment, *Cpi08G004947*-anthocyanidin reductase and other genes were regulated in roots, *Cpi03G002695*-flavanone 3-hydroxylase and other genes were regulated in stems, and *Cpi03G002491*-dihydroflavonol 4-reductase and other genes were regulated in leaves, to influence the accumulation of Cyanidin or Galangin metabolites (Figure 8A and Appendix A). In summary, DEGs orchestrated the biosynthesis of flavonoids in various plant organs of Cp in response to saline-alkali stress conditions.

Analysis of the expression patterns of genes related to the flavonoid pathway revealed that, among different groups, the CK_LeafVSCK_Root group exhibited the highest number of significant DEGs in the flavonoid pathway, with up to 64 genes, as well as the highest number of differential flavonoids (Figure 8A). In NaCl-treated leaves, only four genes showed significant differences, which was consistent with the observation of the lowest number of differential metabolites in the metabolomics analysis for this group (Figure 8A). However, for organs treated with NaHCO_3_, although fewer DEGs were identified in roots, these genes regulated multiple metabolites (Figure 8A). These findings indicated that while the overall trends in gene and metabolite numbers were generally consistent, they did not demonstrate a perfect one-to-one correspondence.

The Venn diagram analysis of genes related to the flavonoid pathway revealed 13 common genes across different organs, with only 1 shared under NaCl treatment and just 2 under NaHCO_3_ treatment, suggesting that distinct gene regulatory mechanisms were employed by different organs under saline–alkali stress, which aligned with the observed organ-specific differences in flavonoids (Figure 8B). Furthermore, when analyzing different organs after NaCl and NaHCO_3_ treatments, it was observed that the number of common genes shared across organs was less than 33% in each case, reflecting that different treatments induced distinct regulatory patterns for flavonoid biosynthesis (Figure 8B).

### 3.5. Differential Transcription Factor Analysis

The aforementioned studies indicated that distinct regulatory mechanisms existed across different organs and saline–alkali stress; so to explore these mechanisms, differentially expressed transcription factors (TFs) were screened from the transcriptomic data. A total of 58 types of differentially expressed TFs were identified, with the top 10 most abundant being bHLH, NAC, ERF, FAR1, C3H, MYB_related, HB-other, B3, WRKY, and C2H2. Specifically, 52 significantly differentially expressed TFs were found in the NaCl_RootVSCK_Root group; 52 in the NaCl_StemVSCK_Stem group; 48 in the NaCl_LeafVSCK_Leaf group; 53 in the NaHCO_3__RootVSCK_Root group; 54 in the NaHCO_3__StemVSCK_Stem group; 54 in the NaHCO_3__LeafVSCK_Leaf group; 58 in the CK_RootVSCK_Leaf group; 57 in the CK_RootVSCK_Stem group; and 57 in the CK_LeafVSCK_Stem group. Given that the number of TF types was relatively similar across different groups and all belonged to the 58 identified categories, it was hypothesized that the differences in TFs among the groups might primarily lie in their quantities. Therefore, the types and quantities of the top 10 most abundant TFs in each group were statistically analyzed (Figure 9A). The results revealed that the differentially expressed TFs among the groups exhibited minimal differences in types but significant variations in quantities.

Transcription factors played crucial roles in plant development and stress responses, while flavonoids also served as important antioxidants in plants. To explore whether TFs synergized with the flavonoid signaling pathway to respond to saline–alkali stress, correlation analysis was conducted between the top differentially expressed TFs in different groups and genes involved in the flavonoid signaling pathway. Due to the large number and variety of TFs, the most abundant TF type in each of the nine groups was selected for correlation analysis. The results revealed that, except for the NaCl_LeafVSCK_Leaf group, where the ERF family was the most numerous, the bHLH family was the most prevalent in the other eight groups (Figure 9A). The correlation analysis between TFs and flavonoid pathway genes indicated a close relationship in all nine groups. Among these, the three groups involving different organs exhibited strong correlations. Additionally, the correlation between these two gene types was relatively high in different organs after saline–alkali stress, with NaCl_RootVSCK_Root having the highest number of correlated pairs (489), while NaCl_LeafVSCK_Leaf had the fewest (only 19 pairs) (Figure 9B). In summary, based on these results, it was speculated that TFs might influence flavonoid synthesis by regulating flavonoid pathway genes.

The above studies demonstrated that TFs were related to flavonoid pathway genes, but whether TFs participated in the synthesis of flavonoids needed further investigation. In order to understand the role of transcription factors in flavonoid synthesis, a correlation analysis between representative *bHLHs* and DAMs of flavonoids in each group was carried out. The results revealed that there was a significant correlation between the *bHLHs* in each group and the metabolites. For example, *Cpi01G01294* in NaCl_LeafVSCK_Leaf was significantly correlated with Cyanidin 3-rutinoside, Luteolin-4′-glucoside, Diosmetin, and Kaempferide; Morin was significantly correlated with 20 genes (*Cpi04G00498*, *Cpi06G004970*, *Cpi02G011870*, *Cpi07G00006*, *Cpi02G010572*, *Cpi05G004115*, *Cpi01G00649*, *Cpi03G006519*, *Cpi02G000602*, *Cpi02G001741*, *Cpi03G005594*, *Cpi02G010428*, *Cpi04G004835*, *C03G002404*, *Cpi02G006273*, *Cpi02G000739*, *Cpi0G000910*, *Cpi01G008793*, *Cpi01G006985*, and *Cpi04G005719*). The above studies indicated that *bHLHs* played an important role in the synthesis of flavonoids (Appendix A).

## 4. Discussion

High-throughput sequencing technology has found extensive application in investigating how plants adapt to saline–alkali conditions [16,17,18]. In our current study, metabolomic and transcriptome analyses were conducted on the roots, stems, and leaves of Cp (showing phenotypic and physiological variations under saline–alkali stress) to elucidate the organ-specific responses of the plant to such stress. And, our findings pinpointed flavonoids and key genes that reacted to saline–alkali stress, thereby aiding us in comprehending the mechanisms underlying the response of different Cp organs to such stress.

When subjected to saline–alkali stress, plants produce detrimental reactive oxygen species (ROS), and an overabundance of ROS can impede plant growth [19]. Our research involved phenotypic observations, which demonstrated that the growth of Cp was suppressed to different extents following saline–alkali exposure, suggesting that various organs of Cp might suffer damage from ROS. To counteract these ROS, plants utilize both enzymatic and non-enzymatic antioxidants as a defensive strategy [20,21,22]. With the aim of exploring the enzymatic antioxidant mechanisms of Cp under saline–alkali stress, we measured antioxidant indices (such as SOD, MDA, GSH, H_2_O_2_, POD, and CAT) in root, stem, and leaf samples that had been treated with 100 mM NaCl and NaHCO_3_ for 6 days. The findings revealed that the same antioxidant index of different organs showed different changes, such as after saline–alkali treatment, the H_2_O_2_ (one of the ROS) content in roots and stems increased while it decreased in leaves, which indicated that appropriate saline–alkali stress was beneficial to the removal of H_2_O_2_ in leaves compared with the CK samples. Moreover, after saline–alkali stress, POD and CAT activities decreased, indicating that the antioxidant system of different organs of Cp was inhibited or antioxidant capacity decreased, resulting in the accumulation of ROS, damage to cells, and ultimately the inhibition of plant growth. Similar observations have also been documented in other studies focusing on different plant organs [23].

In addition to the enzymatic antioxidant system, non-enzymatic antioxidants such as flavonoids also play a pivotal role in the response of Cp to saline–alkali stress. Flavonoids are a class of secondary metabolites widely distributed across various plant organs, enhancing saline–alkali tolerance by scavenging free radicals and through other mechanisms [24,25,26,27,28]. In our study, a total of 32 DAMs of flavonoids were identified, with Cyanidin and Galangin showing different accumulation patterns in different organs of Cp under saline–alkali stress. Among these flavonoids, approximately seven have been reported to be associated with saline–alkali stress, including Naringenin [29], Kaempferide [30], Isorhamnetin [31], Morin [32], Cynaroside [31], Rutin [33], and Cyanidin [34]. Additionally, our research indicated that, compared to roots and stems, leaves contained the highest number of flavonoids, with up to 23. Under NaCl treatment, the numbers of flavonoids in roots, stems, and leaves were 9, 11, and 10, respectively. Under NaHCO_3_ treatment, these numbers were 14, 10, and 11 in roots, stems, and leaves, respectively. These studies demonstrated that various flavonoids underwent alterations in different organs of Cp under saline–alkali stress, and after NaHCO_3_ treatment, the DAMs related to flavonoids in roots were evidently greater than those observed under NaCl treatment, suggesting that these metabolites played significant roles in the saline–alkali response in Cp.

Currently, a diverse array of flavonoid pathway-related genes has been identified in plants. In *Arabidopsis thaliana*, identified genes comprise chalcone synthase (*CHS*), chalcone isomerase (*CHI*), flavanone 3-hydroxylase (*F3H*), flavonoid39-hydroxylase (*F39H*), flavonol synthase, dihydroflavonol reductase (*DFR*), and leucoanthocyanidin dioxygenase [35]. In chrysanthemum, identified genes include *DFR*, *ANS*, *3GT*, *3MaT1*, and *3MaT2* [36]. In our study, compared to roots and stems, leaves contained the most differentially expressed flavonoid-related genes, with up to 54. Under NaCl treatment, the numbers of flavonoids in roots, stems, and leaves were 15, 14, and 4, respectively. Under NaHCO_3_ treatment, these numbers were 9, 17, and 20 in roots, stems, and leaves, respectively. Additionally, these genes include *CHS* genes such as *Cpi01G011650* and *Cpi02G004415*, and *CHI* genes such as *Cpi08G002482* and *Cpi03G000419*, which are involved in the saline–alkali stress response [37]. The aforementioned studies indicated that various genes related to the flavonoid pathway undergo changes in different organs of Cp and under different saline–alkali stress conditions, suggesting that these genes might play crucial roles in the saline–alkali response of different organs in Cp.

In addition, studies have shown that transcription factors also play a crucial regulatory role in plant responses to saline–alkali stress [38,39]. In our study, we found that bHLH transcription factors were the most abundant across all nine groups, and we conducted correlation analyses between them and flavonoid pathway genes, as well as DAMs of flavonoid, revealing significant correlations in each group. And, the roles of bHLH in saline–alkali responses have already been reported, for instance, *Xanthoceras sorbifolia* [5], *Triticum aestivum* [40], and *Asparagus officinalis* [41] respond to saline–alkali stress by regulating transcription factors such as bHLH. Furthermore, the role of these transcription factors in flavonoid biosynthesis has also been documented. For example, the MYB-bHLH-WDR (MBW) complex regulates flavonoid biosynthesis in plants [42]; the genes of the bHLH transcription factor family were the main hub genes regulating flavonoid biosynthesis in *Lycii fructus* [43]; multiple bHLH transcription factors regulate anthocyanin biosynthesis in *Fragaria ananassa* [44]; and ChEGL1 significantly increased the anthocyanin content in transgenic *Cerasus humilis* [45]. In summary, bHLH transcription factors in Cp may respond to saline–alkali stress by regulating flavonoid synthesis.

In conclusion, initially, phenotypic assessments demonstrated that saline–alkali stress induced differential impacts on various organs of Cp. Subsequently, guided by these phenotypic variations, we performed physiological profiling across organs, which uncovered non-uniform changes in antioxidant parameters. These findings imply the existence of organ-specific adaptive strategies to saline–alkali conditions. Ultimately, transcriptomic and metabolomic analyses indicated that flavonoid biosynthetic genes (*CHS* and *CHI*) and bHLH transcription factors (which may regulate the expression of genes such as *CHS* and *CHI*) potentially mediate saline–alkali tolerance by modulating the production of flavonoid compounds, including Cyanidin and Galangin. This study preliminarily deciphers the organ-specific saline–alkali stress response mechanisms in Cp, providing critical insights for the exploitation of flavonoids and facilitating molecular-assisted breeding programs aimed at enhancing stress resilience.

## 5. Conclusions

This study provides the first systematic investigation into the organ-specific saline–alkali stress response mechanisms mediated by flavonoids in Cp. Our integrated phenotypic, physiological, transcriptomic, and metabolomic analyses revealed three key insights: (1) differential organs’ vulnerability to saline–alkali stress, (2) organ-dependent antioxidant adaptation strategies, and (3) molecular regulation of flavonoid biosynthesis via *CHS*, *CHI*, and bHLH transcription factors to enhance stress tolerance. These findings not only advance our understanding of plant–environment interactions in medicinal herbs but also establish a scientific basis for (i) targeted extraction of bioactive flavonoids from specific organs, (ii) molecular-assisted breeding programs to develop salt-tolerant Cp cultivars, and (iii) expanding cultivation into marginal saline–alkali lands. Future research should focus on validating these mechanisms under field conditions and exploring their potential for sustainable agro-medicinal systems in stressed environments.

## Figures and Tables

**Figure 1 biology-14-01759-f001:**
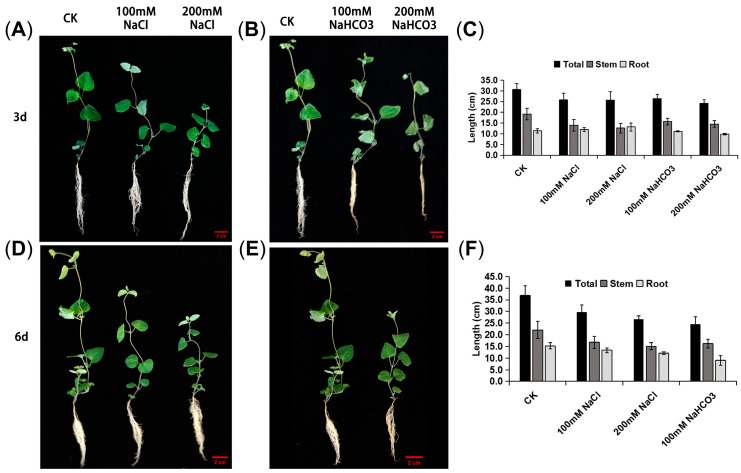
Phenotypic characterization of Cp under NaCl and NaHCO_3_ treatment. (**A**,**B**) Phenotypic images of Cp after 3 days of treatment with NaCl and NaHCO_3_, respectively; (**D**,**E**) phenotypic images of Cp after 6 days of treatment with NaCl and NaHCO_3_, respectively; and (**C**,**F**) denotes the root length, stem length, and total height of Cp after being treated for 3 days and 6 days, respectively (mean ± SD, *n* = 3). Scale bar = 2 cm.

**Figure 2 biology-14-01759-f002:**
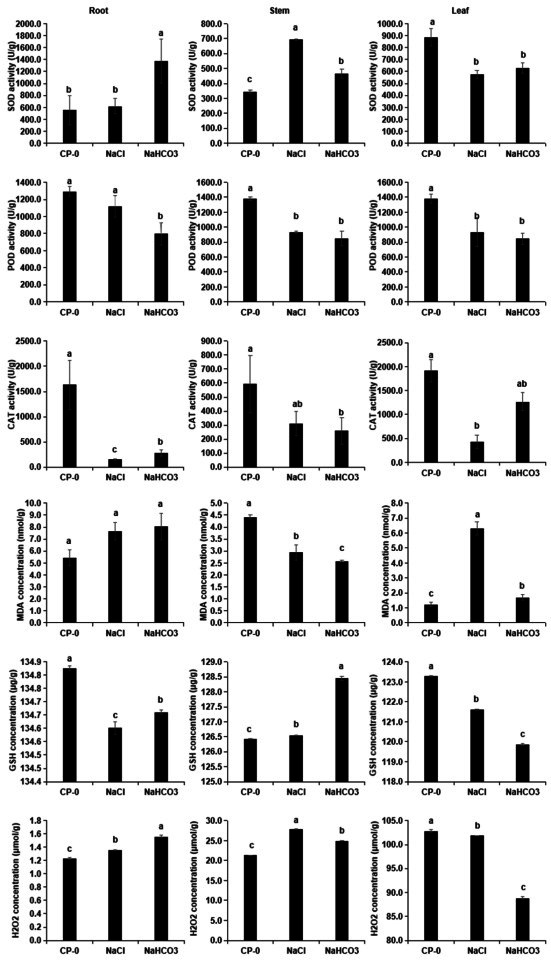
The effects of saline–alkali stress on the antioxidant activity of Cp (mean ± SD, *n* = 3). Columns marked by the same lowercase letters are not significantly different according to Duncan’s multiple range test (*p* < 0.05).

**Figure 3 biology-14-01759-f003:**
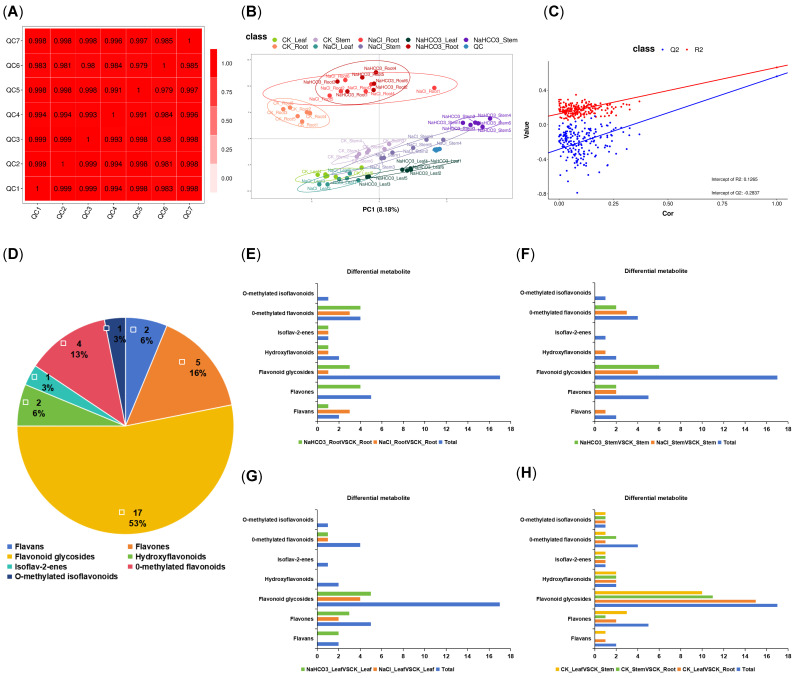
The QC–Pearson correlation diagram and the PLS−DA and quantity distribution diagram of flavonoids. (**A**) QC chart in samples; (**B**) PLS−DA score plot of flavonoids; (**C**) the displacement test chart of flavonoids; (**D**) the classification pie chart of flavonoids; and (**E**–**H**) represent the DAM classification bar charts of flavonoids in roots, stems, leaves under NaCl and NaHCO_3_ treatment, and diffrent orgens, respectively.

**Figure 4 biology-14-01759-f004:**
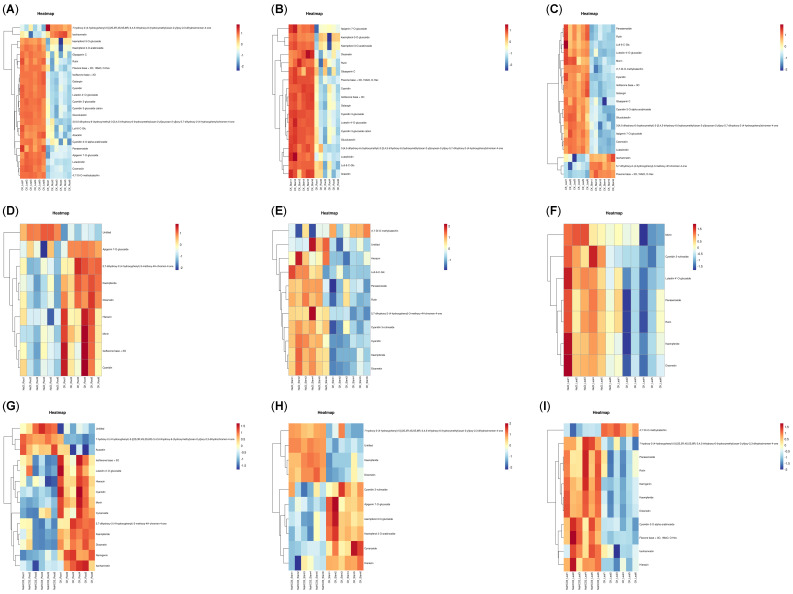
Heatmap DAMs of flavonoids. (**A**–**C**) represents the heatmap DAMs of flavonoids in CK_LeafVSCK_Root, CK_StemVSCK_Root, CK_LeafVSCK_Stem groups; (**D**–**F**) represents the heatmap DAMs of flavonoids in NaCI_RootVSCK_Root, NaCl_StemVSCK_Stem, NaCl_LeafVSCK_Leaf groups; (**G**–**I**) represents the heatmap DAMs of flavonoids in NaHCO_3__RootVSCK_Root, NaHCO_3__StemVSCK_Stem, NaHCO_3__LeafVSCK_Leaf groups.

**Figure 5 biology-14-01759-f005:**
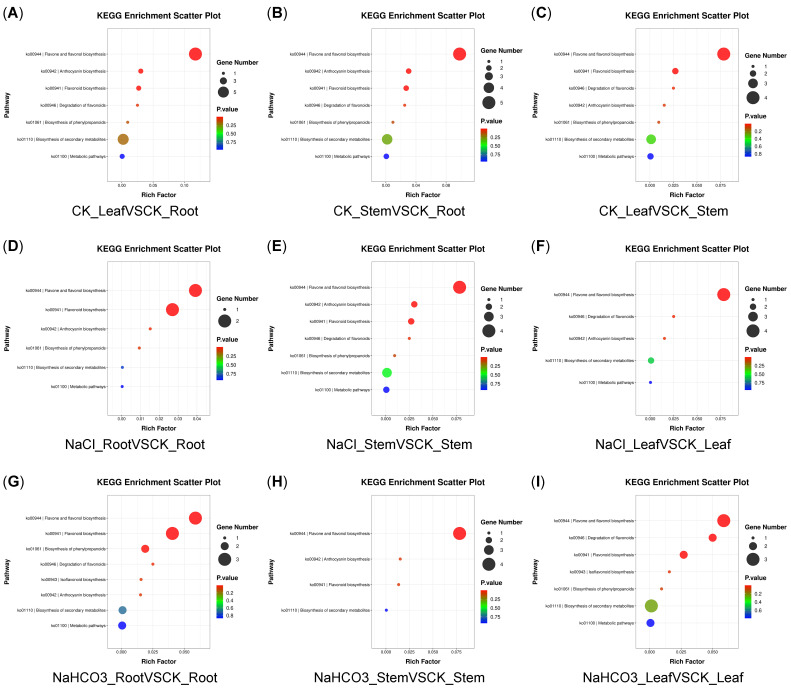
KEGG enrichment analysis map DAMs of flavonoids. The smaller the *p* value, the more significant the enrichment of DAMs.

**Figure 6 biology-14-01759-f006:**
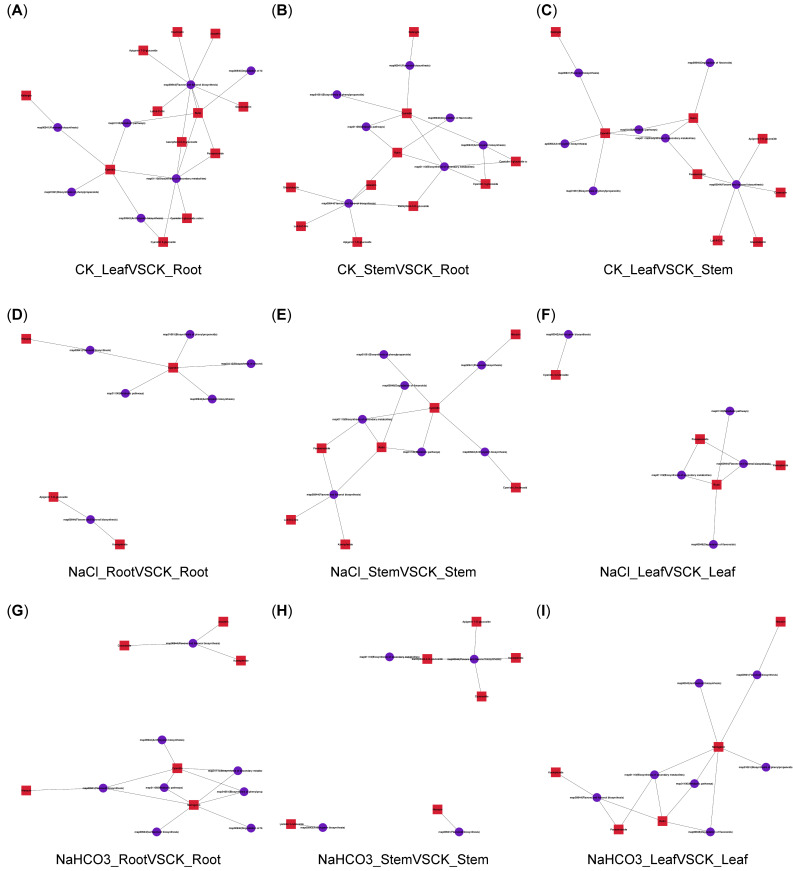
DAMs of flavonoids and the flavonoid synthesis-related pathway correlation network diagram.

**Figure 7 biology-14-01759-f007:**
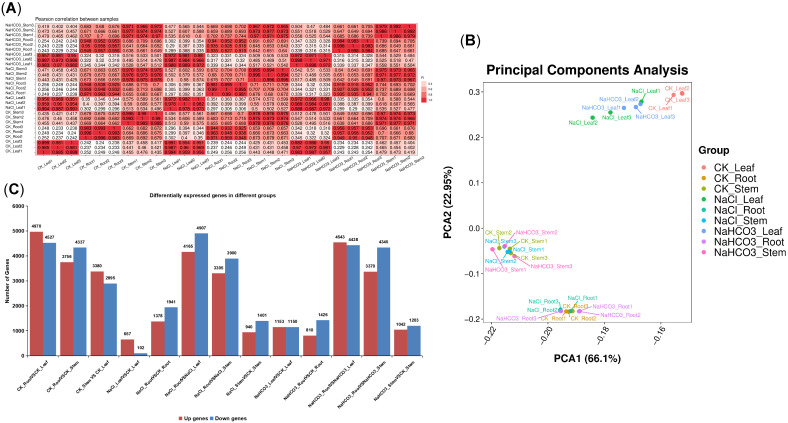
The Pearson correlation coefficient plot, PCA map, and bar chart of the quantity of DEGs. (**A**) Represents the Pearson correlation coefficient plot for different samples; (**B**) represents the PCA map illustrating DEGs among different samples; and (**C**) represents the statistical bar chart depicting the number of DEGs among different groups.

**Figure 8 biology-14-01759-f008:**
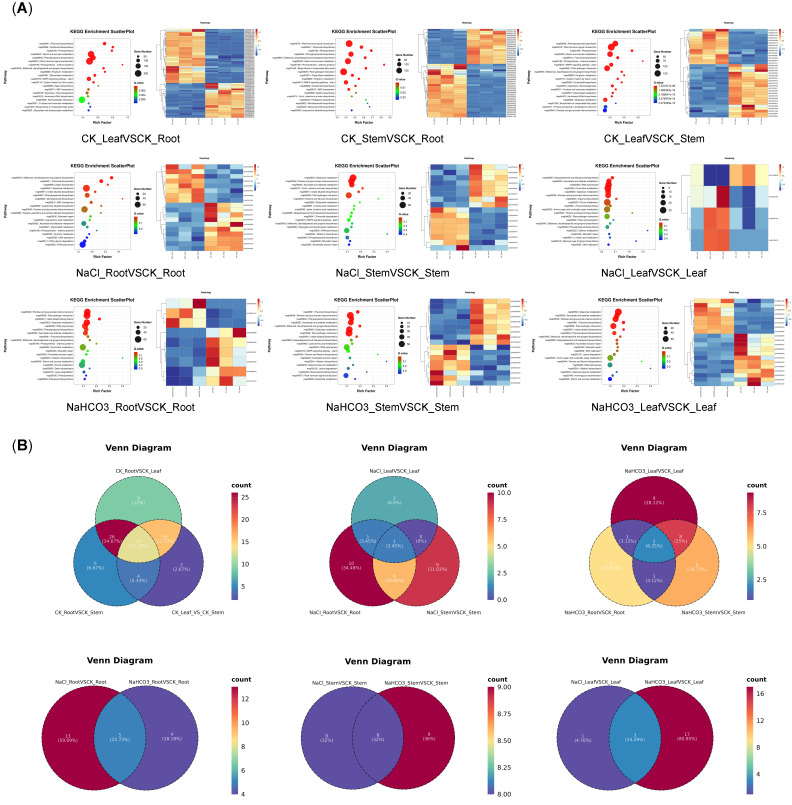
KEGG enrichment analysis diagram, heatmap, and Venn diagram of DEGs in Cp. (**A**) KEGG enrichment analysis and heatmap of DEGs in Cp; (**B**) Venn diagram of DEGs in Cp. The smaller the *p* value, the more significant the enrichment of DEGs.

**Figure 9 biology-14-01759-f009:**
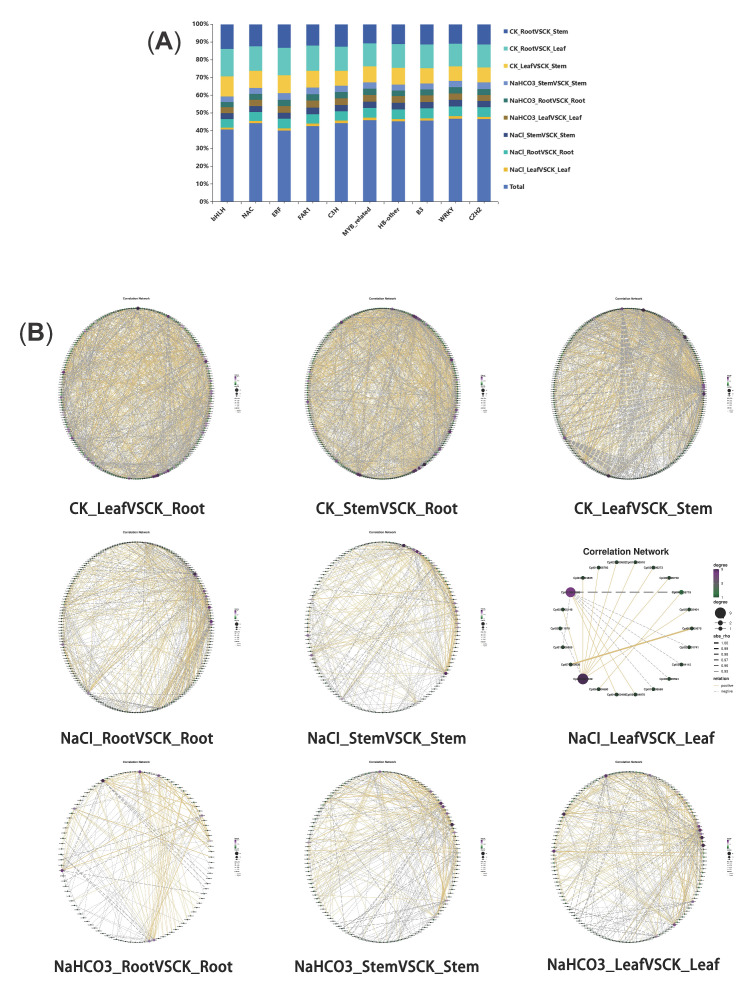
The analysis chart of differential transcription factors. (**A**) Bar chart of differential transcription factors in 9 groups; (**B**) correlation network diagram of bHLH transcription factors in 9 groups,|rho| ≥ 0.9.

## Data Availability

The data used to support the findings of this study are available from the corresponding author upon request.

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
