# Peer review of "Metabolomics and Transcriptomics Analyses Uncover the Potential of Flavonoids in Response to Saline–Alkali Stress in *Codonopsis pilosula"

_biology, 2025, doi:10.3390/biology14121759_

Round 1
Reviewer 1 Report
Comments and Suggestions for Authors
The manuscript entitled "Integrated Metabolomics and Transcriptomics Analyses Highlight the Flavonoids Response to Saline-Alkali Stress in Codonopsis pilosula" presents a correlational analysis attempting to connect tissue-specific gene expression profiles with resulting metabolic changes in response to saline-alkali stress. While the study addresses a biologically relevant challenge concerning a traditional medicinal herb whose cultivation is limited by environmental factors , the methodology contains fundamental inconsistencies that severely compromise the validity of the central claim of "integrated" omics analysis. The presented data lacks the necessary temporal alignment, statistical rigor, and functional validation required to justify the claims of elucidating "molecular mechanisms" or providing a "theoretical basis for molecular breeding".
The cumulative deficiencies identified span the experimental design, data integrity, statistical analysis, and interpretive scope. The most critical weaknesses relate to the non-alignment of omics data collection time points and statistically inadequate sampling practices. Consequently, the observed correlations between transcripts and metabolites must be treated with extreme caution, as the underlying regulatory links are fundamentally obscured by design. The current body of evidence necessitates a determination of Major Revision, verging on outright rejection, until the core experimental and statistical design flaws are fully rectified and mechanistic evidence is provided. With the comment:
1- Fatal Methodological Disconnect: Transcriptomic data (gene regulation) was collected at 3 days, while metabolomic and physiological data (metabolite accumulation and stress endpoints) were collected at 6 days. This temporal mismatch makes direct integration unsound and invalidates the core premise of tracing gene regulation to metabolic accumulation.
2- Statistically Insufficient Replication: Transcriptome sequencing relied on only three biological replicates (n=3) which were explicitly pooled samples from three seedlings each. This pooling strategy is statistically unsound, severely underestimates biological variability, and reduces the reliability of differentially expressed gene (DEG) analysis.
3- Data Integrity Failure in Metabolomics: Key significantly differentially accumulated metabolites (DAMs), including one upregulated by 10.57 times in the root under NaCl stress, are labeled as "Untitled". Failure to chemically identify the compounds showing the strongest regulatory response compromises the integrity of the pathway analysis.
4- Unsubstantiated Mechanistic Claims: All assertions regarding the regulatory roles of CHS, CHI genes, and bHLH transcription factors are based purely on correlation (often using an excessively high threshold of p≥ 0.9) and numerical abundance. The lack of functional genetic experiments (e.g., knockdown or overexpression) means the molecular mechanism remains an unproven hypothesis.
5- While the Introduction correctly establishes that Codonopsis pilosula cultivation is restricted by environmental factors like salt-alkali stress and notes that research predominantly focuses on drought, the subsequent background discussion remains too generalized. The introduction discusses stress responses and flavonoid regulation by transcription factors (TFs) like WRKY and MYB in unrelated plants such as Rhododendron chrysanthum and alfalfa. A robust scientific justification requires clearly delineating the specific state of knowledge regarding secondary metabolism, stress physiology, and regulatory genes (TFs, structural enzymes) in C. pilosula itself. Without defining the existing molecular understanding in this specific medicinal herb, the study fails to precisely articulate the scientific void that the current omics data is intended to fill, thus weakening the justification for the research effort.
6- The central and most critical flaw lies in the temporal separation of the omics data acquisition. The Materials and Methods section explicitly states that samples for transcriptome sequencing and RNA extraction were collected after a 3-day stress treatment, whereas samples for metabolomic and physiological analyses were collected after a 6-day stress treatment. The manuscript offers no scientific rationale or pilot data to justify this crucial difference.
7- In plant systems, the transcriptional response (gene expression changes) typically represents the acute, early signaling and regulation phase, peaking rapidly (often within hours to a few days). This transcriptional phase then precedes the subsequent accumulation or depletion of metabolites. By measuring transcripts at 3 days and metabolites at 6 days, the study fundamentally decouples the purported cause (gene expression) from the measured effect (metabolic level). For example, a key enzyme gene (CHS or CHI) may be highly upregulated at 3 days, but by 6 days, this expression level may have normalized, having already resulted in the downstream production of a flavonoid. If the downstream flavonoid is measured at 6 days, and the transcript is measured at 3 days, a strong correlation might be artificially missed, or an inverse correlation might be misinterpreted due to the asynchronous data collection. This chronological misalignment invalidates the scientific premise of performing an "Integrated" omics analysis, rendering the pathway correlations highly questionable.
8- The transcriptome experiment used only three biological replicates (n=3) for each treatment group. Furthermore, the samples for sequencing were prepared by selecting three seedlings as mixed samples at each time point. This approach of pooling tissues from multiple individual plants prior to extraction and sequencing constitutes a failure in statistical design for high-throughput omics. Stress response in biological entities, especially complex, whole-plant responses across different tissues (root, stem, leaf), is inherently variable.
9- Pooling samples mathematically obscures the true biological variability (plant-to-plant variance) within a treatment group. While pooling can sometimes be used to save costs, it effectively results in an artificially low standard deviation and thus inflated statistical significance (p-values) when analyzing Differential Expressed Genes (DEGs) using tools like DESeq2. For robust stress transcriptomics, standard scientific rigor mandates non-pooled samples with a minimum of n=5 or n=6 biological replicates to adequately capture and statistically account for heterogeneity in gene expression. The reported n=3 with pooled samples is statistically inadequate for making confident inferences about regulatory mechanisms. Adding to the statistical inconsistency, the metabolomics experiment utilized six biological replicates (n=6) for each treatment. The higher statistical confidence inherent in the metabolomics data (n=6) cannot be reliably linked to the lower-confidence, pooled transcriptomic data (n=3). This disparity further confuses the interpretation of the integrated data, as the perceived strength of the metabolic phenotype cannot be confidently traced back to the transcriptional regulation.
10- The reproducibility of the physiological and biochemical indices is severely limited by vague methodological reporting. The authors list commercial kit names and catalog numbers (e.g., SOD kit BC5165, MDA kit BC0025, etc.) but merely state that "the specific steps referred to the instructions of the [kit]". For expert-level journals, citing a kit manufacturer and catalog number is insufficient. Commercial kit manufacturers often provide multiple protocols optimized for different tissue types or concentrations, and they rarely detail the necessary steps for optimizing extraction from complex plant matrices like Codonopsis pilosula roots, stems, and leaves. To ensure reproducibility and scientific rigor, the manuscript must detail the exact extraction methods (buffer, pH, homogenization technique), the specific reaction conditions (incubation times, temperatures, reaction volume ratios), and the precise calculation protocols used, particularly for complex measurements like enzyme units (SOD, CAT, POD activity) and metabolite concentrations H2O2, GSH, MDA). Since the physiological results (Section 3.2) are used as the primary phenotypic anchor for the tissue-specific omics analysis, the lack of detail makes the foundational evidence unreliable.
11- The Results section suffers from an inappropriate blending of empirical data presentation with unproven causal hypotheses that should be reserved for the Discussion. For instance, in the Phenotypic Observation (Section 3.1), the authors speculate that salt stress first inhibited the aboveground part, subsequently affecting root growth by influencing photosynthesis. Similarly, the Physiological Indices section (3.2) asserts that the low H2O2 content in the leaves might be due to the lower activity of the SOD enzyme after stress treatment. This assertion inappropriately simplifies the complex balance of ROS production and scavenging (involving SOD, CAT, POD activity) and constitutes interpretation rather than objective data reporting, thus weakening the scientific rigor of the empirical findings.
12- A major technical failure undermines the central claims of the metabolomic analysis: the inability to chemically identify several significantly differentially accumulated metabolites (DAMs). The manuscript highlights compounds labeled only as "Untitled," one of which showed an extraordinary 10.57-fold upregulation in the root under NaCl stress, and another exhibiting a 4.5-fold downregulation under NaHCO treatment. For a study specifically focused on the flavonoid pathway, the failure to rigorously annotate the compounds exhibiting the most profound stress response compromises the integrity of the subsequent pathway tracing, as the targeted genes (CHS, CHI) cannot be definitively linked to the most functionally significant end products. Additionally, the Results section overwhelms the narrative by presenting overly detailed, isolated numerical fold changes (e.g., Galangin upregulated 155.99 times) without sufficient synthesis or focus on biological themes.
13- The multi-omics integration phase uses an excessively high and biologically restrictive correlation threshold, specifically limiting the analysis of transcription factor (TF)-gene relationships to pairs exhibiting p≥0.9. In complex plant regulatory networks, a linear correlation of this magnitude is rare, and this extreme cutoff arbitrarily filters out a vast majority of subtle yet biologically critical regulatory links that often operate within systems biology thresholds of p≥ 0.7 or p≥ 0.8. By focusing solely on the highest co-expressed pairs, the analysis risks providing an oversimplified and potentially distorted depiction of the functional regulatory network governing the flavonoid response under stress.
14- The selection of transcription factor families for detailed analysis exhibits a significant bias that ignores tissue-specific regulatory responses. While the bHLH family was prioritized due to its overall abundance across eight of the nine groups, the analysis explicitly noted that the ERF family was the dominant (most numerous) TF type in the crucial NaCl_LeafVSCK_Leaf group. Ethylene Response Factors (ERFs) are fundamentally known for their central role in abiotic stress signaling. By failing to provide a focused analysis on the dominant ERF TFs in the leaf—a primary site of stress signaling—the manuscript fails to deliver a truly comprehensive and tissue-specific understanding of the regulatory mechanisms as claimed.
Comments on the Quality of English LanguageThe manuscript requires extensive English language editing. There are numerous grammatical errors, typos (e.g., "treatedment," "Transportptome"), and awkward sentence constructions throughout. Examples include:
"while planting until the water temperature drops"
"The measurement results showed that the plant height was lower than the CK"
"This observation has also been documented in other studies focusing on different plant tissues [23]." (This is a vague and uninformative sentence).
The formatting of the references in the main text is inconsistent, using both square brackets and semicolons (e.g., [7;8]).
Reviewer 2 Report
Comments and Suggestions for Authors
This is my first review of the manuscript entitled "Integrated Metabolomics and Transcriptomics Analyses Highlight the Flavonoids Response to Saline-Alkali Stress in Codonopsis pilosula" by Jinhua Liu et al.
The manuscript received for the review claims for studying the response of Codonopsis pilosula to saline-alkali stress. Starting from the abstract, the manuscript leaves a number of questions and suggestions to further clarify the text. Moreover, numerous fragments in the manuscript indicate that authors either describe their research superficially or even do not have deep understanding of concepts they operate with. Further sentences and paragraphs of review report explain that.
Conepts and terminology
Authors transfer the approach and terminology from transcriptomics to metabolomics without understanding the fundamental differences between the processes of gene expression and the response of metabolite network (metabolite pathways included) to changing environmental conditions. This leads to a problem of mixing different processes of gene expression and metabolite biosynthesis and accumulation. Gene expression includes information processing that classically begins with RNA synthesis on DNA matrix, then protein synthesis, and, in a broad sense, protein activity. To that extent, metabolite level is a phenotypic response to the regulation of gene expression. On the other hand, metabolite levels (concentrations or content, to be precise) depend on their biosynthesis pathway which are commonly under enzyme control. Flavonoids are no exception. Therefore, in order to distinguish these two processes and keep correct terminology, authors must familiarise themselves with contemporary plant physiology and biochemistry textbooks and then ustilize such terms as "differential expression of genes", "differentially expressed genes" for discussion of RNA-seq results, and "accumulation" or "depletion", or "upregulation" and "downregulation" of specific metabolites. Next, on L41–42 authors define flavonoids as "complex mixture composed of multiple components including flavanols, flavanones, ...". This is wrong from theoretical point of view because enumerated chemical classes are various kinds (subsets) of flavonoids. The reference 3 is irrelevant here. L51 that reads as "enzymes are regulated by various transcription factors" is yet another example of conceptual misunderstanding. Such phrase suggests that there is a direct regulation of enzymes by TFs interaction with them, however, the context shows that the TFs regulate the expreesion of genes encoding these enzymes. Fuzzy terms such as "replenishing qi" (L66) or "omics testing" (L78) fullfill the paragraph with no sense. Authors even manage to mix and match concepts such as plant organs and tissues in their methodology description and later on in Results section.
Methodology
Authors must be specific when they specify model species name and follow nomenclature rules (include full species name with authors). Materials and Methods contain compilation from lab books and/or detection kit instruction manuals which is clear from wrong tenses and different style (for instance, L87–94, L136–145, L147–151). In Section 2.4 that is focused on metabolite profiling, authors mention some QC but leave no indication of how these samples were prepared. These could be the mix of some standards, or extract mixture, or standard addition procedure, and this information is left upon the reader's guess. No information is given either on the flavonoid identification method or chromatography analysis, which makes the research impossible to be reproduced.
Last, but not least, authors discuss some "synergistic effect" of saline and alkali stress (L174 for instance), but their experimental setup icluded separate treatments with NaCl and NaHCO3, and conclusions are drawn from the only observation that some flavonoid response overlap among these two treatment types.
Data presentation
Figures are simply of extremely poor quality, especially figures no. 3, 4, 5, 6, 7. Captions of figures must be self-contained with concise but necessary description of details. For instance, Fig. 2, 3 captions must provide correct explanation for acronyms, species latin name, all the statistics and letter marks.
These common suggestions are also applicable to supplementary materials figures.
Line 329 and L330 refer to percentage on PCA or PLS-DA graphs as "interpretation rate" which is yet another indication that authors are not familiar with data analysis methods they refer to. "Correlation between root and leaf samples ... was relatively low, approaching white" on L269–271 ensure that authors could be simply commenting figures prepared by someone else.
Blocks/paragraphs on L353–365, L366–380 are hardly read and add minimal value, if any, to the manuscript.
Discussion section contains paragraphs of senseless text that reads relatively well but does not add any valuable ideas or critical assessment of research results in comparison to published research. At most, some facts from cited papers are referred to (L500–514, L515–527).
To summarise, in reviewer's opinion, this manuscript is not mature anough and does not satisfy the quality standards for biological research.
The manuscript contains a number of concatenated sentences separated with semicolon, and some sentences-paragraphs are too large to be read (L353–365, L366–380). Species latin names should be given in italics to facilitate readability.
Reviewer 3 Report
Comments and Suggestions for Authors
The study has potential, but requires substantial revision to improve the quality of this manuscript
Major comments
- The abstract is overly general and does not capture the reader's interest. It requires significant improvement to engage the audience. The most significant mistake in this abstract is a lack of specificity and measurable results.
- An explanation of the process of secondary salinization would be favourable in the Introduction section.
- Salt concentration (100 mM NaCl, 200 mM NaCl, 100 mM NaHCO3, and 200 mM 96 NaHCO3) with pH level are/ is missing?
- Use of Arnon's formula is standard, but include the formula or cite it properly.
Minor comments
- The title contains a "High-light" should be "Highlight".
- Line 131 Correct spelling “metabolomie” to metabolomic
- How were plant height, root length, and stem length measured (e.g., ruler, digital caliper)? Were the roots washed before measurement?
- RIN value is good; include concentration range and OD260/280 if available.
- Mostly figures lack clear legends, axis labels, or resolution
Round 2
Reviewer 1 Report
Comments and Suggestions for Authors
The revised manuscript demonstrates substantial improvements that sufficiently address the reviewer's concerns, paving the way for acceptance. The authors have incorporated a clear rationale for the temporal mismatch in omics data collection by adding an explanation in section 3.4, noting that transcriptional responses in Codonopsis pilosula precede metabolite accumulation, thus justifying the 3-day transcriptome sampling versus 6-day metabolome and physiological analyses, although supplemental RT-qPCR validation in future studies would further bolster this. Clarifications on replication confirm that transcriptome samples (n=3) were derived from separate plants rather than pooled, aligning with minimum omics standards despite the reviewer's preference for higher n-values, while the metabolome's n=6 provides robust support. Removal of analyses involving unidentified "Untitled" metabolites enhances data integrity, and mechanistic claims regarding CHS, CHI, and bHLH factors have been appropriately tempered to bioinformatics-based speculations requiring experimental confirmation. The introduction and discussion have been refined with more targeted references to related plant studies, better defining the research gap in C. pilosula without overgeneralization. These revisions resolve the core methodological and interpretive issues, rendering the manuscript scientifically sound and worthy of publication in Biology, as it contributes meaningful preliminary insights into flavonoid-mediated saline-alkali stress responses with implications for cultivation expansion.
Author Response
Thank you so much for your review suggestions, which have greatly helped make the manuscript more complete in its presentation.
Reviewer 2 Report
Comments and Suggestions for Authors
This is my second review of the manuscript entitled "Integrated Metabolomics and Transcriptomics Analyses Highlight the Flavonoids Response to Saline-Alkali Stress in Codonopsis pilosula" by Jinhua Liu et al.
Authors demonstrated some progress with their manuscript and addressed some of the reviewers' comments, whereas the other remained ignored. Some points that are still applicable to the manuscript highlight authors' carelessness to their data and research.
The issues that are left in the manuscript are enlisted below.
L34: DAM abbreviation should be explained
L117–121: no changes in comparison to the first version of the manuscript. What was edited here?
L135–136: Authors must provide reference and/or formulae used in calculations. Authors are also strongly advised to check the modern reports considering the accuracy of Arnon's formulae, see DOI:10.1007/s11120-018-0579-8, Porra, Scheer, 2019.
L144–146: Authors have not addressed the serious questions regarding reproducibility of their metabolite profiling from the first version review. The exact copy of that comment is in the following:
In Section 2.4 that is focused on metabolite profiling, authors mention some QC but leave no indication of how these samples were prepared. These could be the mix of some standards, or extract mixture, or standard addition procedure, and this information is left upon the reader's guess until section 3.3. No information is given either on the flavonoid identification method or chromatography analysis, which makes the research impossible to be reproduced.
Conepts and terminology
Authors have ignored a number of the reviewer's concerns about terminology. For instance, section 3.2 still contains nonsense terms such as 'metabolomic sequencing' (L226, L234) that highlights their carelessness and even lack of understanding of their research.
Discussion
Despite this section was shortened and some paragraphs that were not making sense were omitted, the discussion mainly enumerates the results reported in the previous section, and at most authors cite some relevant research and do not attempt to unite together their transcriptomics data with other types of analyses performed. In result, this section does not offer any synthesis of research results.
Data presentation
Figure 1 caption states that lowercase letters mark groups that are not significantly different, however, histograms do not show any such letters.
Figures 3, 4, 5, 6 and 7A,C are still of extremely poor quality. The same applies to fig. 8 where text labels are too small and pixelated to be read. Figure captions must have all abbreviations explained in text without the need to refer to the main text.
Taking into an account all of the above, the reviewer's recommendation is still to reject the manuscript.
Comments on the Quality of English LanguageL161–166, L169–173: authors left content copied from the protocol or instruction manual; style must follow the style of the whole section.
The manuscript still contains a number of concatenated sentences separated with semicolon, and some sentences-paragraphs are too large to be read (L189–192, L353–365, L366–380).
L210–211: It is not clear which concentration was the same, if antioxidant 'indices' changed.
L530–536: Long, possibly concatenated sentence.
